# Swine Gut Lactic Acid Bacteria and Their Exopolysaccharides Differentially Modulate Toll-like Receptor Signaling Depending on the Agave Fructans Used as a Carbon Source

**DOI:** 10.3390/ani15071047

**Published:** 2025-04-04

**Authors:** Enrique A. Sanhueza-Carrera, Cynthia Fernández-Lainez, César Castro-De la Mora, Daniel Ortega-Álvarez, Claudia Mendoza-Camacho, Jesús Manuel Cortéz-Sánchez, Beatriz Pérez-Guillé, Paul de Vos, Gabriel López-Velázquez

**Affiliations:** 1Laboratorio de Biomoléculas y Salud Infantil, Instituto Nacional de Pediatría, Ciudad de Mexico 04530, Mexico; enriquesanhueza@comunidad.unam.mx; 2Posgrado en Ciencias Biológicas, Universidad Nacional Autónoma de México, Cuidad de Mexico 04510, Mexico; 3Laboratorio de Errores Innatos del Metabolismo y Tamiz, Instituto Nacional de Pediatría, Ciudad de Mexico 04530, Mexico; lainezcynthia@ciencias.unam.mx; 4Layan Biotic Solutions, Guadalajara 44670, Mexico; cesarcastro@bggroup.mx (C.C.-D.l.M.); danielortega8605@gmail.com (D.O.-Á.); claudia.m@bggroup.mx (C.M.-C.); 5Centro de Enseñanza, Investigación y Extensión en Producción Porcina, FMVZ–UNAM, Jilotepec 54240, Mexico; manuelcortez@fmvz.unam.mx; 6Translational Research Center, Instituto Nacional de Pediatría, Ciudad de Mexico 04530, Mexico; bettyepg@yahoo.com; 7Immunoendocrinology, Division of Medical Biology, Department of Pathology and Medical Biology, University and Medical Center of Groningen, 9700 Groningen, The Netherlands; p.de.vos@umcg.nl

**Keywords:** lactobacilli, swine, agave, fructans, probiotics, exopolysaccharides, TLRs

## Abstract

Pork is an essential source of animal protein for humans, and swine production is an important component of agricultural economics and trade. However, the presence of swine diseases still represents a significant challenge. Regarding swine healthcare, we aimed to investigate whether using three different graminan-type fructans from agave plants as carbon sources for swine lactic acid bacteria would show immunomodulatory potential to enhance consumers’ health. The exopolysaccharides that these lactic acid bacteria strains deliver into the environment were assayed to assess their immune properties. Since exopolysaccharides’ physical and chemical composition are essential for their biological function, we observed a carbon source dependence in regard to the immunomodulatory effects of strains of lactic acid bacteria from swine and the exopolysaccharides that they produce. The exopolysaccharides produced by lactic acid bacteria grown with the shorter graminan-type fructan showed the best effects in regard to reducing inflammation. The characteristics of the proposed synbiotic model are beneficial in promoting a healthy intestine and enhancing the immune response in swine or other mammals. Since exopolysaccharides are generally regarded as safe products, the model studied here could be safely applied to enhance swine health and ensure successful farming.

## 1. Introduction

From an anthropocentric point of view, swine are essential in several ways. For instance, swine, rather than rodents, are a better translational animal model in biomedical research, due to their similarity to humans in terms of their anatomy, physiology, and genetics [1,2]. The overlap between human and swine microbiomes is high at the functional level, showing a closer similarity between swine and human gut microbiomes than between mouse and human gut microbiomes [3].

Additionally, pork meat is an excellent source of protein and is the most widely consumed food worldwide [4]. However, significant swine losses, due to infections in production animals, lead to substantial economic losses for farmers. To cope with these challenges, probiotics are used in swine production to improve performance, mitigate diseases, increase product quality, and reduce environmental pollutants [5].

It is important to emphasize that alterations in the intestinal microbiota and immune response in production animals create favorable conditions for pathogenic bacteria, enhancing their colonizing ability and exacerbating gut inflammation [6]. Such an inflammatory state impairs the intestinal epithelium’s integrity, weakening the gut barrier, causing more permeability, and further escalating inflammatory processes [7]. In the search for strategies to cope with these immunity-related diseases, probiotics, prebiotics, and postbiotics have been commonly used to modulate gut health and enhance the immune response and the effectiveness of vaccination [8,9,10,11,12].

Various bacterial genera have been used as probiotics in swine; however, lactic acid bacteria (LAB) members are the most widely used probiotic agents [13]. Additionally, mixtures of prebiotics and probiotics have demonstrated that they can support vaccination efficacy against multidrug-resistant pathogens [14]. Such a combination of probiotics and prebiotics (synbiotics) has synergistically beneficial effects on host health [15,16]. Additionally, postbiotics, such as bacterial exopolysaccharides (EPSs), are becoming more relevant in enhancing the health of humans and animals. EPSs are high molecular weight sugar-based biopolymers produced by various bacteria (and other microorganisms), whose quality and functional properties strongly influence the carbon source used during bacterial growth [17,18,19].

Carbohydrates, such as fructans, derived from plants, are good candidates for use as carbon sources in developing functional probiotics and EPSs to enhance health in various animals, including swine. This is mainly due to their immunomodulatory, anti-inflammatory, and prebiotic properties [20,21,22,23]. Graminan-type fructans (GTFs) derived from agave plants are of interest, since mounting evidence indicates that their health benefits depend primarily on the structure of the individual polysaccharides and oligosaccharides in such mixtures [24]. GTFs consist of fructofuranosyl units, featuring a mixture of β (2→1) and β (2→6) linkages with a terminal glucose residue [25], resulting in branched molecules with high solubility that are colorless, odorless, and tasteless.

The use of GTFs as carbon sources for the growth of probiotic bacteria favors the synthesis of metabolites, sugars, and micronutrients, such as short-chain fatty acids (SCFAs), which improve local or systemic immunity against pathogens [26,27,28]. Our group recently demonstrated that GTFs from the *Agave tequilana* Weber Blue variety directly affected the immune response through nuclear factor kappa B and the activator protein 1 pathway (NF-κB/AP1), opening a new field of research into their effects on health [29,30,31]. Mexican agaves are enriched in GTFs, and their structural characteristics depend on the species from which they are obtained [32]. Therefore, this study aimed to investigate the direct immunomodulatory effects through the NF-κB/AP1 pathway of lactic acid bacteria (LAB) isolated from swine feces and cultured with three different agave GTFs as carbon sources. Additionally, the isolated EPSs produced by these LABs in the abovementioned conditions were assayed to evaluate their direct immunomodulatory effects through extracellular toll-like receptors. Herein, we demonstrate the strong influence of GTFs on LAB’s immunomodulatory effects, illustrating its specificity and its impact on the EPS structure and its biological effects.

Based on our results, we highlight that the potential use of this system, which combines GTFs and LAB as a supplement in pig production, will promote weight gain and increase the average daily feed intake and feed-to-gain ratio, as reported by others using similar systems based on probiotics and prebiotics [33]. The impact on pig production will include improved performance during pregnancy, parturition, and lactation in sows, enhancing immunohematological parameters and defenses during the growing phase, influencing meat quality during the finishing phase, and helping to reduce environmental pollutants. Additionally, it has been demonstrated that feeding piglets growth promoters, based on prebiotics and probiotics, results in better economic benefits. This supplemented feeding reduces mortality, lowers the costs of veterinary treatment, increases weight gain, and improves feed conversion, resulting in a net economic benefit of 1:7.5 compared to animals without supplementation [34,35]. Consequently, the economic benefits of implementing this kind of supplemented feeding in the pig industry are worth considering.

## 2. Materials and Methods

### 2.1. Ethical Statement

All of the procedures carried out as part of this study were approved by the Institutional Research Committee and the Committee for the Care and Use of Laboratory Animals (IACUC-INP) under project register code INP 2021/026. The Institutional Research Committee is certified by the Federal Commission for the Protection against Sanitary Risks (COFEPRIS, Mexico), register number 17 CI 09 003 109.

### 2.2. Animals

The minipigs (*Sus scrofa domesticus*) used in this study belong to a local genetic line developed by a breeder through the selective crossbreeding of small pigs from various breeds. The primary selection included pigs with white skin and others with white coats, featuring black or brown spots. At birth, their average weight is 250 g, and by two years of age, they reach a body weight ranging between 25 and 35 kg. The minipigs were obtained from RGS Research Global Solutions (RGS07090347A, Mexico City, Mexico).

Since the colony’s establishment in 2015, all the animals have been housed in a purpose built, conventionally managed, and ventilated facility, designed explicitly for minipigs. The facility complies with the NOM 062 ZOO 1999 guidelines (https://www.gob.mx/cms/uploads/attachment/file/203498/NOM-062-ZOO-1999_220801.pdf, accessed on 2 October 2024). The location of the establishment ensures biosecurity, as no other pig farms are located within a 20 km radius. Additionally, a dedicated caretaker is responsible for all the animals, who does not work with other swine populations, and follows strict protective measures, including wearing specialized clothing when entering the unit.

The animals were provided with untreated tap water and received a restricted diet consisting of Purina^®^ (Mexico City, Mexico) feed (https://www.llabana.com/jamonina-pt, accessed on 15 December 2024). The feeding regimen consisted of 200 g of feed per meal, twice daily, at 9:00 a.m. and 5:00 p.m., with the feed placed directly on the floor. Purina ham does not contain agave GTFs. However, it contains 6.5% fiber, composed of subproducts of cereals, such as wheat bran, corn bran, and cane molasses. In contrast, GTFs are known to be present in wheat grains [36].

Sample monitoring is carried out once a year, with the most recent sampling taking place in June 2024. Serum samples were taken from five animals to test for viral and bacterial antibodies. Samples were also taken for parasitological examination, showing that the swine were free of Aujeszky’s disease, circovirus, porcine epidemic diarrhea, PRRS, *Actinobacillus pleuropneumoniae*, *Pasteurella* spp., *Bordetella bronchiseptica*, *coccidia* (Eimeria, Isospora), and *helminths*.

The animals underwent a one-month acclimatization period in the animal facility, adhering to the institution’s internal regulations. The facility meets the standards outlined in the Guide for the Care and Use of Laboratory Animals (National Research Council) and is AAALAC accredited, ensuring compliance with the NOM-062-ZOO-1999 regulations. The housing conditions included pens with slotted concrete floors, nipple drinkers, a controlled temperature range of 22–26 °C, and a relative humidity of 50% ± 20%. The pen sizes are designed to meet the space requirements for minipigs, providing at least 1.08 m^2^ per animal. Each pig was housed in a pen measuring 1.12 m × 1.82 m.

The minipigs received daily socialization and participated in a standardized enrichment program to promote their well-being and behavioral enrichment. Their health and welfare were monitored twice daily to ensure optimal conditions throughout the study.

### 2.3. Fecal Sample Collection

Fecal samples were collected from five healthy minipigs (four males and one female), from the animal facility at the National Institute of Pediatrics, Mexico City, Mexico, with the abovementioned characteristics. The pigs were in the growing stage, with their body weight ranging from 17.6 kg to 24.6 kg. They were all 9 months of age and fed Purina^®^ ham, as described above. Pig fecal samples were collected directly from fresh stools, using sterile swabs. The samples were placed in sterile containers and stored individually under refrigeration until processing for lactic acid bacteria isolation, which occurred within one hour after collection. The animals were not subjected to any experimental procedures. However, the collection of fecal samples was performed in compliance with the guidelines for animal research.

### 2.4. Isolation of Lactic Acid Bacteria (LAB) Strains

In regard to the LAB isolation procedure, 1 g of feces from each animal was used in aseptic conditions. A tenfold dilution series was made using 1 g of feces in 0.85% sterilized saline, homogenized with a sterile swab, until the solid content was homogenized. Subsequently, 0.1 mL of the serial dilutions (10^−1^–10^−8^) were plated on Man–Rogosa–Sharpe (MRS, Difco, MD, USA) agar plates, supplemented with 0.3% aniline blue to identify the LAB [37]. The plates were incubated at 37 °C and 5% CO_2_ for 72 h. After incubation, the total number of LAB was determined by selecting colonies with typical LAB characteristics [38]. Each typical LAB colony was transferred into tubes containing 10 mL of MRS broth. The isolated LAB were sub-cultured on MRS agar plates and incubated at 37 °C and 5% CO_2_ for 72 h. To characterize the isolated colonies as LAB, they were subjected to biochemical tests, such as the Gram test and assessments in regard to its sporulating capacity and catalase activity. The isolated colonies were preserved in cryotubes and stored at −70 °C in a 50% (*w*/*v*) sterile glycerol solution for further studies.

### 2.5. Graminan-Type Fructans

The carbon sources of agave fructans (GTFs) for LAB growth were GTF A, Fiber Prime™ (Guadalajara, Mexico); GTF B, Flora Advantage™ (Guadalajara, Mexico); and GTF C, Fiber Balance™ (Guadalajara, Mexico), which were all isolated from Mexican agave plants (Industrializadora de Fructanos Tierra Blanca™, Guadalajara, Mexico). GTF A is a mixture composed of 76.36% ± 0.02 fructans, with a degree of polymerization (DP) > 10 and a smaller proportion (17.26% ± 0.01) of shorter chains, with a DP ranging from 3 to 10. GTF B has an equal proportion of long chains with a DP > 10 and short chains with a DP of 3–10 (43.46% ± 8.24). GTF C is a mixture of 54.99% ± 0.16 short chains, with a DP of 3–10, and 33.26% ± 0.03 long chains, with a DP > 10. The manufacturer provided the DPs of the GTFs.

### 2.6. Determination of Growth Curves of LAB

To evaluate which GTF could function as the best carbon source for growing LAB, growth curves were derived using GraphPad™ Prism software (version 10.4.1, Boston, MA, USA), based on a turbidity measurement, as indicated by an increased OD_600 nm_ value [39]. A fresh 10 mL overnight MRS culture was prepared for each LAB strain under study, and the cell density was determined using a spectrophotometer (Varian, Cary 50, Santa Clara, CA, USA). For strain inoculation, the number of cells was adjusted at a final OD_600 nm_ of 0.1. For the culture media with different carbon sources, the bacteria were washed three times in phosphate-buffered saline (PBS) and then resuspended in 1 mL of MRS medium under the following conditions: MRS with 10 g/L of glucose as the control for basal EPS production, MRS–GTF A, MRS–GTF B, or MRS–GTF C under aerobic conditions at 37 °C and 5% CO_2_. The OD _600 nm_ was measured every 2 h (for the first 12 h), followed by aliquots taken at 18 h, 20 h, 24 h, 36 h, 48 h, and 50 h. All the samples were tested in triplicate, and the averages were calculated and plotted.

### 2.7. Selection Criteria for LAB Strains

To maintain the LAB strain that produced the highest amounts of EPSs, independent cultures were prepared in 10 mL of sterile MRS broth from the six LAB-isolated colonies and were cultured at 37 °C and 5% CO_2_ for 48 h. Afterward, the EPSs were extracted and quantified from the supernatant, as detailed in the following section.

### 2.8. Extraction and Quantification of EPSs from the LAB Culture Supernatant

With some modifications, the method described by Ferrari et al. was followed for EPS extraction [40]. Bacterial cells were harvested using centrifugation (4500 rpm, 4 °C, 30 min). Then, the resulting supernatant was treated with trichloroacetic acid (TCA) at a final concentration of 10% (*v*/*v*) and incubated at 4 °C for 24 h. After centrifugation (10,000 rpm, 4 °C, 30 min), the supernatant was separated from the protein precipitate, and two volumes of 95% ethanol (EtOH) were added to the supernatant. The resulting mixture was incubated at 4 °C, overnight. Pellets were collected through centrifugation (10,000 rpm, 4 °C, 30 min), resuspended in sterile distilled water, and dialyzed (3.5 kDa MWCO) in 1 L of distilled water for 24 h, with two changes of the water per day. The total carbohydrate content was quantified using the phenol–sulfuric acid method [41], employing GTFs from Fiber Prime™ as a standard for constructing a calibration curve (10 μg/mL to 200 μg/mL).

### 2.9. EPS Production by LAB Using Agave GTFs as Carbon Sources

A modified carbohydrate-free MRS medium (cfMRS) was prepared [42]. This formulation contained: 7 g of meat extract, 7 g of peptone, 5 g of yeast extract, 3 g of sodium acetate × 3 H_2_O, 2 g of K_2_HPO_4_, 1.2 g of NH_4_Cl, 0.2 g of MgSO_4_ × 7 H_2_O, and 0.06 g of MnCl_2_ × 4 H_2_O. The pH was adjusted to 6.5 using 1N HCl. As the primary carbon source for fermentation, three media based on the above formulation were supplemented with 25 g/L of GTF A (MRS–A), GTF B (MRS–B), and GTF C (MRS–C). Each control strain and the LAB strain under study were cultured separately in the three MRS media with the different GTFs at 37 °C and 5% CO_2_ for 48 h. For a comparison of the EPS production, we included two ATCC LAB strains: *Limosilactobacillus reuteri* strain F275 [DSM 20016], ATCC 23272; and *Lacticaseibacillus casei* strain 03, ATCC 393. The EPSs were then extracted from the supernatant for quantification, as described in Section 2.8.

### 2.10. Assays with Reporter Cell Lines

To evaluate whether the LAB that used GTFs as carbon sources and their EPSs activated or inhibited the NF-κB/AP1 pathway, a THP1 reporter cell line was used (Invivogen, Toulouse, France). This cell line is derived from the human THP1 monocyte cell line, which endogenously expresses all human TLRs. It has been genetically modified for the stable integration of an NF-κB-inducible SEAP (secreted embryonic alkaline phosphatase) reporter construct. Therefore, when environmental stimuli activate TLRs, the activation of the NF-κB pathway can be quantified through SEAP production, which can be spectrophotometrically registered at 655 nm. Cells cultured with only the RPMI medium were considered as negative controls. The positive controls were cells challenged with 10 ng/mL of *Escherichia coli* lipopolysaccharide (LPS) (Table 1). Additionally, human embryonic kidney cells (HEK-Blue™, Invivogen, Toulouse, France), which independently express TLR2 and TLR4, were used. These cell lines are also genetically modified for SEAP production induced by the NF-κB pathway promoter. The cell densities, the agonists used as positive controls, and the selection antibiotics are shown in Table 1.

The THP1 and HEK-Blue cell lines were cultured in RPMI-1640 and DMEM media, respectively. Both media were supplemented with 10% deactivated fetal bovine serum, 100 μg/mL of Normocin, and 50 U/mL and 50 μg/mL of penicillin/streptomycin, respectively. According to the manufacturer’s instructions, the cell lines were passaged twice weekly and used at 80% confluency.

### 2.11. Statistical Analyses

The normality of the data was determined using the Shapiro–Wilk test. For the normally distributed data, a one-way ANOVA followed by Dunnett’s multiple comparisons adjustment was used for the significance analysis. The Mann–Whitney U or Friedman test was used for the non-parametric distributed data, followed by Dunn’s multiple comparisons adjustment test. The results were expressed as the mean ± SD or the median and interquartile range (IQR) for data with parametric and non-parametric distributions, respectively. A *p*-value < 0.05 was considered to be statistically significant (* *p* < 0.05, ** *p* < 0.01, *** *p* < 0.001, **** *p* < 0.0001). At least six independent experiments, each one with three technical replicates, were performed for each experiment. All these analyses were performed using GraphPad Prism™ software (version 10.4.1 (532), San Diego, CA, USA).

## 3. Results

### 3.1. LAB Strains Isolated from Swine Feces and Their EPS Production

A total of six LAB strains were isolated and characterized as Gram-positive bacilli, non-sporulating, and catalase negative. They were INP_MX_001, INP_MX_002, INP_MX_003, INP_MX_004, INP_MX_005, and INP_MX_006. We sampled the total number of individuals in the herd (five): four males and one female. However, we isolated one LAB strain from each individual, except for one individual (male #1), from which two strains were isolated, resulting in a total of six isolated LAB strains (Appendix A). As an initial approach, the six strains were cultured in MRS medium, and their EPSs were isolated and quantified (Figure 1). The INP_MX_001 strain emerged as the best EPS producer of all six isolated strains.

### 3.2. The LAB Strain Growth Was GTF Dependent

Since the INP_MX_001 LAB strain was the best EPS producer from the six isolated LAB strains (Figure 1), it was selected for further analyses. The tested INP_MX_001 LAB strain showed good growth enhancement, with differences depending on the GTF used. Moreover, the viability of the LAB strain cultured with GTFs was constant during the 50 h of the experiment (Figure 2).

### 3.3. The LAB Strain Isolated from Swine Was the Best EPS Producer with GTFs as a Carbon Source Compared with Other Well-Characterized Lactobacilli

The capacity for EPS production by the INP_MX_001 LAB strain grown with three different GTFs as carbon sources was investigated. This finding was compared with the capacity of two well-characterized species of lactobacilli, *Lacticaseibacillus casei*, and *Limosilactobacillus reuteri*, to produce EPSs, also grown with the same carbon sources. To that end, all the LAB strains were grown with the three GTFs, followed by the extraction and quantification of their EPSs. The INP_MX_001 LAB strain was the best EPS producer compared with the other lactobacilli assayed. This finding was very GTF dependent since, when grown with GTF A, it produced 308.6 ± 1.7 μg/mL of EPSs. Meanwhile, *Lacticaseibacillus casei* and *Limosilactobacillus reuteri* only produced 152.2 ± 5.3 μg/mL and 141.2 ± 2.3 μg/mL, respectively, with the same carbon source (Figure 3). The production of EPSs by the INP_MX_001 LAB strain was the best when GTF B and C were used as the carbon sources, although in smaller amounts compared with GTF A, since it produced 102.8 ± 2.6 μg/mL and 106.8 ± 1.9 μg/mL, respectively. Furthermore, independently of the carbon source, the amounts of EPSs produced by the INP_MX_001 LAB strain were higher compared with the lactobacilli species assayed (Figure 3).

### 3.4. LAB Activate the NF-κB Pathway Independently of the Carbon Source, but Its Activation Is EPS Dependent

The activation of the NF-κB pathway by the LAB strain was investigated. First, THP1 reporter cells were incubated for 24 h with 1.16 × 10^9^ CFU/mL of the LAB previously grown with the different GTFs as carbon sources, followed by the quantification of SEAP production. This cell line is derived from the human THP-1 monocyte cell line through the stable integration of an NF-κB-inducible SEAP (secreted embryonic alkaline phosphatase) reporter construct. We found that the LAB grown with GTF C as a carbon source was the most potent dose-dependent activator of the NF-κB pathway, with a range of activation from 6.62 ± 0.24 to 11.33 ± 0.23 (*p* < 0.0001). The LAB grown with GTF A and GTF B as carbon sources similarly and significantly activated the NF-κB pathway (Figure 4A).

We also evaluated whether the EPSs produced by the selected LAB strain grown with the different GTFs could activate the NF-κB pathway in THP1 reporter cells. When the cells were incubated with the different EPSs, we found that all of them could activate the NF-κB pathway in a dose-dependent manner. The EPSs from GTF A were the strongest activators of the pathway. All of the tested concentrations significantly activated the pathway, with activation ranging from 2.63 ± 0.42 to 18.08 ± 0.83-fold compared with the untreated control (*p* < 0.0001). This differed for the EPSs from GTF C, whose activation ranged from 2.76 ± 0.42 to 8.41 ± 0.61-fold (*p* < 0.0001). The weakest activator of the NF-κB pathway was the EPS from GTF B, since only the highest concentration significantly activated the pathway, with a 2.59 ± 0.41-fold increase (*p* < 0.0001) (Figure 4B).

### 3.5. The Activation of TLR2 by LAB Is Carbon Source Dependent; Conversely, EPSs Produced by LAB Independently Activate TLR2

The THP1-Blue™ cell line is highly responsive to pattern recognition receptor (PRR) agonists that trigger the NF-κB pathway, including agonists for TLR2, TLR1/2, and TLR2/6 interactions with their cognate ligands, followed by TLR4, TLR5, and TLR8. At the same time, responses to TLR3, TLR7, and TLR9 are hardly detectable. To dissect the contribution of extracellular TLRs to the activation of the NF-κB pathway observed in THP1 reporter cells, we incubated TLR2 reporter cells in the presence of the selected LAB strain. In another set of experiments, we incubated the cells in the presence of the EPSs extracted from the LAB grown with different GTFs as carbon sources. Compared with the untreated control, we found that the LAB strain that was grown with GTF C was the only one that significantly activated TLR2 in the range from 1.45 ± 0.11 to 2.32 ± 0.38 (*p* < 0.0001) (Figure 5A). When incubating the cells with the EPSs, the activation of TLR2 was significantly activated, independent of the carbon source; however, only the highest concentration of EPSs from GTF B activated TLR2, with a 1.52-fold ± 0.27 increase (*p* < 0.01), while the highest concentrations of the EPSs from GTF A activated TLR2, with increases of 1.62 ± 0.29-fold (*p* < 0.001) and 1.58 ± 0.42-fold (*p* < 0.001), respectively. All the tested EPS concentrations from GTF C significantly activated TLR2, with activation ranging from 1.40 ± 0.34-fold (*p* < 0.05) to 1.48 ± 0.31-fold ± (*p* < 0.01) (Figure 5B).

### 3.6. LAB Strongly Inhibit TLR2 Activation, While EPSs Produced by LAB Inhibit It in a Carbon Source-Dependent Manner

The observed effect of the selected LAB strain and the EPSs on TLR2 could depend on their activating and inhibitory activities. Therefore, we investigated their inhibitory effects. First, TLR2 reporter cells were pre-incubated for 1 h with 1.16 × 10^9^ CFU/mL of the selected LAB strain and with 1:10, 1:100, and 1:1000 serial dilutions, followed by the addition of the TLR2 agonist, Pam3CSK4, and 24 h of incubation. Compared with Pam3CSK4, and independent of the carbon source, a strong and significant inhibitory effect on TLR2 activation was observed with the undiluted LAB strain and all of its tested dilutions. The inhibitory range was from 0.05 ± 0.007-fold (*p* < 0.0001) from the LAB strain grown with GTF A as a carbon source to 0.35 ± 0.13 (*p* < 0.0001) from the 1:1000 dilution of the LAB strain grown with the same GTF (Figure 6A). This was different for the EPSs. The inhibitory effect of the EPSs was significant in all the tested conditions, but was carbon source dependent. The highest inhibition was observed with the EPSs obtained when the LAB strain was grown with GTF C as a carbon source, ranging from 0.29 ± 0.06-fold (*p* < 0.0001) to 0.43 ± 0.08-fold (*p* < 0.0001) (Figure 6B).

### 3.7. The Selected LAB Strain Grown with All the GTFs Except GTF A Activated TLR4, and the EPSs from GTF A Activated It in a Dose-Dependent Fashion

To further analyze the TLR-activating effect observed in THP1 cells, the TLR4-activating capacity of the selected LAB strain grown with three different GTFs as carbon sources was studied in the HEK-Blue–TLR4 reporter cell line. These cells were incubated with our LAB strain and the produced EPSs, followed by SEAP quantification. Compared with the untreated control, we observed that the LAB strain grown with GTF B and GTF C significantly activated the NF-κB pathway via TLR4, while the LAB strain grown with GTF A did not activate TLR4. The LAB strain grown with GTF C activated TLR4, in a range from 1.9 ± 0.26-fold to 2.8 ± 0.77-fold (*p* < 0.0001). This was different for the LAB strain grown with GTF B, which activated TLR4 to a lesser extent, with activation ranging from 5.3 ± 0.28-fold (*p* < 0.001) to 6.8 ± 0.67-fold (*p* < 0.0001) (Figure 7A). When testing the TLR4-activating capacity of the EPSs produced by the selected LAB strain grown with the different GTFs as carbon sources, we found that, compared with the untreated control, the EPSs from GTF A were the strongest TLR4 activator, with activation ranging from 6.0 ± 0.54-fold (*p* < 0.0001) to 8.6 ± 2.4-fold (*p* < 0.0001). The highest concentration of EPSs from GTF B was the only one that activated TLR4, with a 4.2 ± 0.85-fold increase (*p* < 0.0001). The weakest TLR4 activator was the EPS from GTF C, with activation ranging from 1.8 ± 0.13 (*p* < 0.05) to 2.1 ± 0.6-fold (*p* < 0.01) (Figure 7B).

### 3.8. The Inhibition of TLR4 Activation by the LAB Strain Is Carbon Source and Dose Dependent, While Its EPSs Inhibit It Independently of These Variables

We tested the TLR4 inhibitory capacity of the selected LAB strain grown with the three different GTFs as carbon sources and their produced EPSs. First, HEK-Blue TLR4 cells were pre-incubated with the LAB and the EPSs for 1 h, followed by the addition of the TLR4 agonist LPS and an extra 24 h incubation. The inhibitory capacity of TLR4 increased as the dilution of the LAB strain decreased in all cases (Figure 8A). The most potent TLR4 inhibitor was the LAB from GTF B, ranging from 0.59 ± 0.14-fold (*p* < 0.0001) to 0.82 ± 0.13-fold (*p* < 0.001). Independent of the carbon source or the concentration, the EPSs from GTF A and GTF C significantly inhibited TLR4 activation, and the most concentrated condition of the EPSs from GTF B was the strongest inhibitor, with a 0.66 ± 0.06-fold decrease (*p* < 0.0001) (Figure 8B).

## 4. Discussion

The health benefits of LAB and their EPSs are well-known in humans and animals [43,44]; however, their immunomodulatory effects have rarely been studied. In the present study, we isolated a LAB strain from swine feces and tested whether these bacteria and their produced EPSs could signal via TLRs. To the best of our knowledge, this is the first study to demonstrate the immunomodulatory effect via TLRs of a LAB strain isolated from swine feces and of its EPSs, using three structurally different GTFs from agave plants as carbon sources.

The TLR-activating capacity of the selected LAB strain was investigated. First, we incubated the reporter cell line, THP1, in the presence of different concentrations of the LAB strain grown with different GTFs as carbon sources, followed by the quantification of the SEAP production. We found that the LAB strain significantly activated the TLRs, independent of the carbon source. We also questioned whether the EPSs produced by the selected LAB strain were able to stimulate THP1 cells. After incubating them in the presence of EPSs, we found that they activated TLRs in a dose-dependent manner. This is similar to what our group found in a previous report, where THP1 cells were incubated in the presence of 16 different health-promoting LAB strains isolated from several human foods and feces. Significant activation of the NF-κB pathway was observed in regard to most of the tested strains [45]. However, this might be a bacterial species-dependent effect since, in another recent report from our group, we did not find any activation of TLRs in THP1 cells by EPSs isolated from *Bifidobacterium longum* subsp. infantis strain DSM 20088 and *Bifidobacterium adolescentis* strain DSM 20083 [46,47].

In light of our results, activating the NF-κB pathway through the whole LAB cell is likely not carbon source dependent in terms of the culturing system. THP1 cells (which express all the TLRs typically expressed in mammalian cells) are activated similarly and independent of the LAB cell’s concentration (see Figure 4A). On the other hand, the EPSs isolated from these LAB clearly activated the THP1 cells in a dose-dependent manner and differentially in terms of magnitude, depending on the carbon source in the culturing system (see Figure 4B). Additionally, the most substantial fold change in terms of activation was with the EPSs from the LAB cultured in the presence of GTF A (17 times higher than the highest activation obtained with LAB cells). This supports the finding that EPS activation of the NF-κB pathway strongly depends on the carbon source in the culturing system, whereas this does not apply in regard to LAB. However, further studies must be carried out to understand the underlying mechanisms of such behavior.

Since our isolated LAB strain and its EPSs were able to activate TLRs in THP1 reporter cells, we wondered whether TLR2 and TLR4 on the cell surface are involved in such an effect. Therefore, reporter HEK-Blue cell lines that independently express these receptors were incubated with different concentrations of the LAB strain that was grown with three different GTFs as carbon sources. In a different set of experiments, these cells were also incubated with increasing concentrations of the EPSs produced by the LAB mentioned above, followed by the quantification of the SEAP production. We found that the LAB strain grown with GTF C as a carbon source was the only one that activated TLR2. This could be a chain length-dependent effect of the carbon source, since GTF C is composed mainly of short chains, with a DP that ranges from 3 to 10. Other authors, such as Murofushi et al., have previously described the activation of TLR2 by LAB strains [48]. They incubated transfected cells expressing TLR2 in the presence of a strain of *L. plantarum*, N14, or its EPSs and found that the NF-κB pathway was activated [48]. In another report by Wang et al. 2019, it was found that a strain of *L. plantarum* derived from swine modulated the synthesis of an intestinal host defense peptide and that this was dependent on the TLR2/MAPK/AP-1 signaling pathway [49].

Our study found that all the EPSs produced by the isolated swine LAB strain, independent of the carbon source used, activated TLR2 in a dose-dependent manner. This is similar to the TLR2-activating capacity of the EPSs extracted from the biofilm of the YT-1 strain of *Thermus aquaticus* [50]. The activation of TLR2 is achieved when it forms heterodimers with TLR1 or TLR6. This ultimately leads to the activation of immune response genes, including the production of pro-inflammatory cytokines via the transcription factor, NF-κB.

NF-κB signaling through TLR4 was not activated by the whole bacteria of the LAB strain grown with GTF A. However, the LAB grown with GTF B and C activated TLR4, independent of the concentration. In contrast, the activation induced by EPSs from LAB grown with GTF A was significantly strong and dose dependent, suggesting a high sensitivity of TLR4 to these EPSs. Similar results were obtained by Laplanche et al., who tested several strains of *Ruminococcus gnavus* and their EPSs [51]. Both their results and ours suggest that the shedding of these molecules in the external milieu may play a role in LAB–host immunomodulation. It has been proposed that this may be due to the glucose substitution pattern of EPSs, i.e., the higher the glucose substitution, the lower the recognition of EPSs by TLR4. However, it is worth mentioning that EPSs from *R. gnavus* that highly activate TLR4 appear less inflammatory in eliciting IL-12p40, IL-6, and TNF-α compared to EPSs, which hardly activate TLR4 signaling at all [51]. This suggests that other PRRs may be involved in the host response to this type of EPS. The observed activation of TLR4 by the whole cells of the LAB strain that was grown with GTF B and GTF C as carbon sources could be related to the structures adopted by their capsular polysaccharides (CPSs), which are influenced by the carbon source used to grow them. CPSs are tightly linked to the cell surface, often covalently, whereas EPSs are secreted into the extracellular medium. A recent study by Wang et al. found that CPSs from *Bacteroides fragilis* ameliorate the abnormal metabolism of the fungal agent, voriconazole, via TLR4 signaling [52]. Since the detailed structural characteristics of the EPSs we obtained are still absent, further studies must be conducted to corroborate this idea.

From the inhibition analyses, we found that, independent of the carbon source present during their growth, the LAB strain could inhibit the activation of TLR4. This is similar to what was seen in regard to *B. fragilis* by Wang et al. [52]. Moreover, the produced EPSs also significantly inhibited TLR4 signaling, independent of the GTF used during their growth. Such TLR4 inhibitory properties are known to lead to health benefits and have been reported for EPSs from probiotics such as *Lactobacillus plantarum* [53], *Lacticaseibacillus rhamnosus* [54], *Limosilactobacillus mucosae* [55], and *Bacillus subtilis* [56,57], to name but a few.

It is worth noting that the whole cells of the LAB strain showed strong inhibitory effects on the TLR2 signaling independent of the GTF used as a carbon source. Their EPSs also exerted significant inhibitory effects on TLR2. However, the inhibition through EPSs depended on the source of GTF used during LAB growth. The shorter the size of the GTF used as a carbon source, the stronger the inhibitory effect of the EPSs on TLR2 signaling. Our group previously reported TLR2 signaling inhibition by EPSs from *Bifidobacterium infantis* and *B. adolescentis* without GTFs as a carbon source [46]. However, those inhibitory effects were not as strong as those observed here, involving the LAB grown with GTF C. This is because *B. infantis* and *B. adolescentis* EPSs differential TLR2 inhibition might be attributed to the higher galactose (GAL) content in the latter. Our results likely present similar but enhanced behavior, as previously proposed by Galván-Moroyoqui et al. [58]. Our results support the finding that the EPSs of the isolated LAB grown with GTFs might reduce inflammation and prevent excessive activation of immune receptors, such as by lowering TLR signaling. Hence, the EPSs from LAB grown with GTF C (the shorter GTF) would have the best effects on reducing inflammation. These characteristics of the proposed model are beneficial in promoting a healthy intestine and enhanced immune response in swine and other mammals.

Although TLR2 and TLR4 activate the NF-κB pathway to a comparable extent, they exhibit notable differences in their impact on cytokine and chemokine gene transcription. This disparity indicates that the signals generated by these TLRs are far from identical. Moreover, the distinct roles of their pattern recognition receptors may significantly influence the polarization of adaptive immunity, underscoring their importance in immune response modulation. Cytokine production through TLR4 activation is associated with type I immunity (Th1) responses, whereas TLR2 activation produces cytokines that favor Th2 development [59]. Th1 activates cell types that are actively involved in controlling infections by intracellular pathogens, while Th2 stimulates antibody production to control infections by extracellular pathogens [60]. Therefore, assaying the activation with TLR2 and TLR4 cell lines helped us to explore the potential activation of the immunity type that could exert the different LAB and EPS conditions explored herein. Since detailed analyses of the produced cytokines are still lacking, we cannot precisely conclude what kind of pathogens (intracellular or extracellular) are better targets for the LAB and EPS systems studied in this work.

Our findings on the immunomodulant features of the LAB model grown with different GTFs as carbon sources underscore that it is a promising model for reducing infections and that it may be used to enhance swine survival as livestock. Additionally, as previously demonstrated for isomalto/maltoopolysaccharides [61], our model could also counteract the immune-attenuating effects of several antibiotics by activating the TLRs 2 and 4 on the cell surface and the associated NF-κB signaling pathway. Given that EPSs are biological macromolecules with unique physicochemical attributes and rheological properties with a generally regarded as safe (GRAS) status, the model studied here could be safely applied to enhance health in mammals, such as swine or even humans. Further studies must be performed to demonstrate the usefulness of this model in vivo. As future directions in terms of the present study, further studies will focus on the chemical and structural characterization of the GTFs used as carbon sources, the EPSs isolated from the selected LAB strain, their structure–function correlations, and the short-chain fatty acids that the LAB produce with the different carbon sources.

## 5. Conclusions

Taken together, the findings in the present work suggest that the LAB strain we isolated from swine, grown with three structurally different GTFs as carbon sources, as well as its produced EPSs as effector molecules, may have health benefits in swine (or another host), which could be exploited in the farming and livestock industry.

## Figures and Tables

**Figure 1 animals-15-01047-f001:**
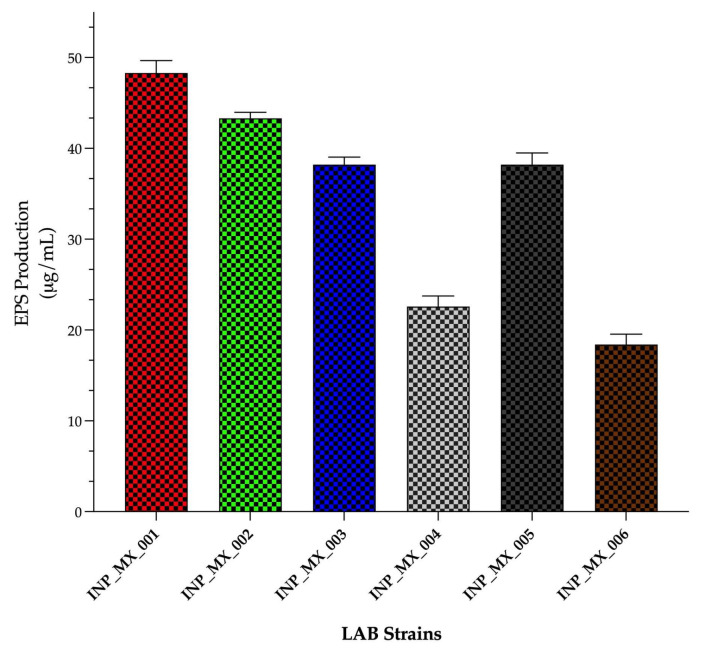
Exopolysaccharide production by LAB strains isolated from swine feces. Data are expressed as EPS micrograms isolated from each milliliter of LAB culture.

**Figure 2 animals-15-01047-f002:**
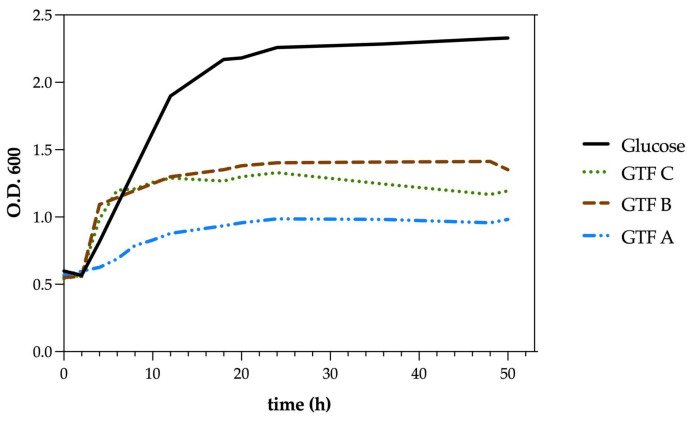
Growth curves of the INP_MX_001 LAB strain incubated with three different graminan-type fructans as carbon sources.

**Figure 3 animals-15-01047-f003:**
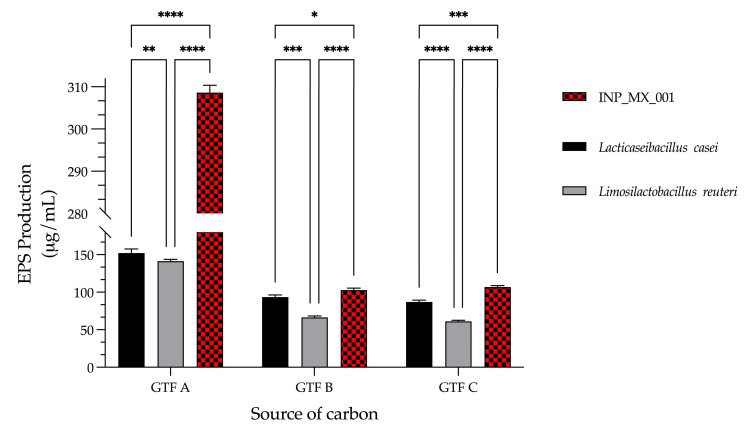
Comparison between the EPSs produced by the LAB strain isolated from swine feces and the EPSs produced by other well-characterized lactobacilli incubated with three different graminan-type fructans as carbon sources. The data are expressed as EPS micrograms isolated from each milliliter of bacterium culture. * *p* < 0.05, ** *p* < 0.01, *** *p* < 0.001, **** *p* < 0.0001.

**Figure 4 animals-15-01047-f004:**
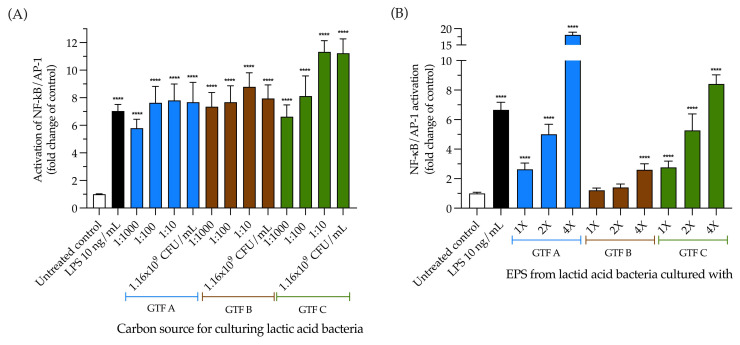
Activation of the NF-κB pathway in THP1 reporter cells by LAB grown with different GTFs as carbon sources and by EPSs obtained from the abovementioned LAB. (**A**) Effects of the LAB strain when incubated using the different GTFs as carbon sources; (**B**) effects of the EPSs extracted from the abovementioned LAB strain under the same conditions. Activation is shown as the fold change of the untreated control. The results are plotted as the mean ± Std. deviation; **** *p*-value < 0.0001.

**Figure 5 animals-15-01047-f005:**
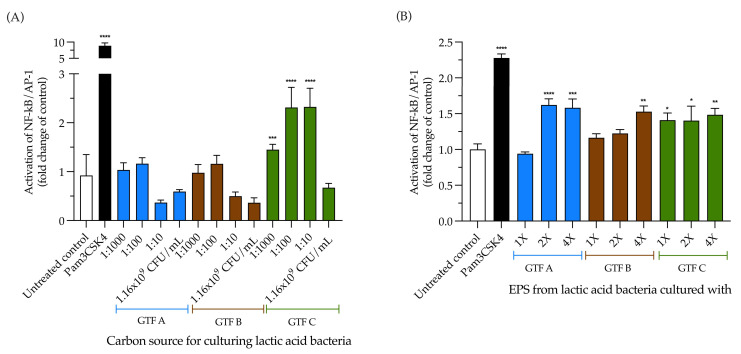
The activation of the NF-κB pathway via TLR2 by LAB grown with different GTFs as a carbon source and by EPSs was obtained from the LAB mentioned above. (**A**) Effects of the LAB strain when incubated using the different GTFs as carbon sources; (**B**) effects of the EPSs extracted from the abovementioned LAB strain under the same conditions. Activation is shown as the fold change of the untreated control. The results are plotted as the mean ± Std. deviation; * *p* < 0.05, ** *p* < 0.01, *** *p* < 0.001, **** *p* < 0.0001.

**Figure 6 animals-15-01047-f006:**
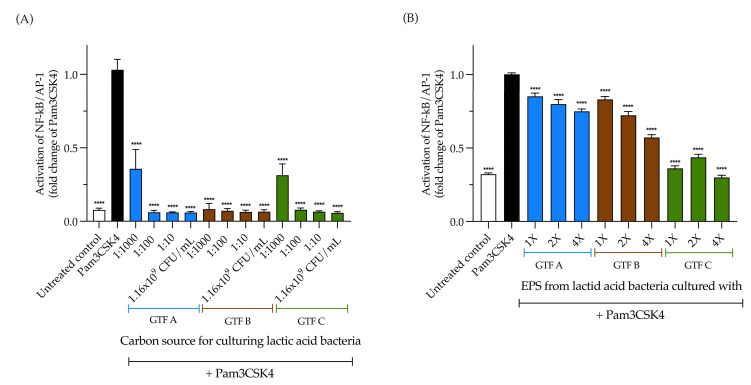
Inhibitory effect of a LAB strain isolated from swine and grown with three structurally different graminan-type fructans (GTF A, B, and C) and inhibitory effect of EPSs obtained from this LAB strain on the TLR2 reporter cell line. (**A**) Effects of the LAB strain when incubated using the different GTFs as carbon sources; (**B**) effects of the EPSs extracted from the abovementioned LAB strain under the same conditions. Inhibition is shown as the fold change of the TLR2 agonist, Pam3CSK4. The results are plotted as the mean ± Std. deviation; **** *p* < 0.0001.

**Figure 7 animals-15-01047-f007:**
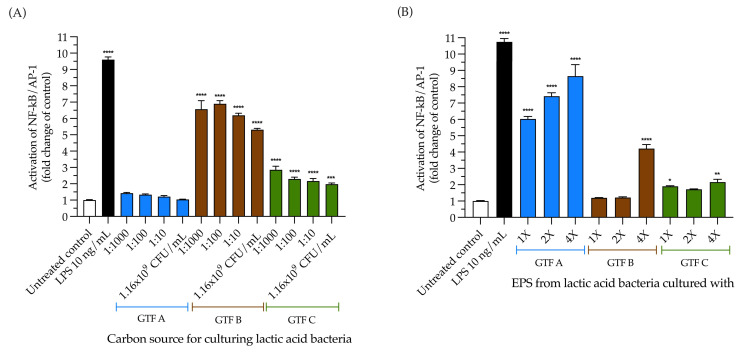
Activation of NF-κB pathway via TLR4 by LAB grown with different GTFs as carbon sources and by EPSs obtained from the LAB mentioned above. (**A**) Effects of the LAB strain when incubated using the different GTFs as carbon sources; (**B**) effects of the EPSs extracted from the abovementioned LAB strain under the same conditions. Activation is shown as the fold change of the untreated control. The results are plotted as the mean ± Std. deviation; * *p* < 0.05, ** *p* < 0.01, *** *p* < 0.001, **** *p* < 0.0001.

**Figure 8 animals-15-01047-f008:**
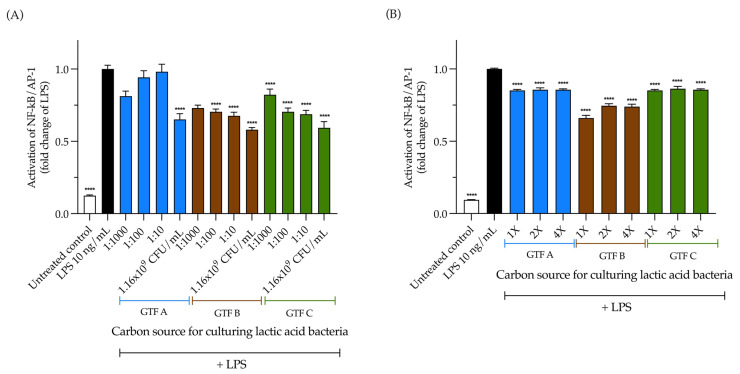
Inhibitory effects of a LAB strain isolated from a pig, grown with three different graminan-type fructans (GTF A, B, and C) obtained from three GTFs, and inhibitory effects of EPSs obtained from this LAB strain on the TLR4 reporter cell line. (**A**) Effects of the LAB strain when incubated using the different GTFs as carbon sources; (**B**) effects of the EPSs extracted from the abovementioned LAB strain under the same conditions. Inhibition is shown as the fold change of the TLR4 agonist, LPS, 10 ng/mL. The results are plotted as the mean ± Std. deviation; **** *p* < 0.0001.

**Table 1 animals-15-01047-t001:** Densities of reporter cell lines, agonists, and selection markers used for reporter cell lines.

Cell Line	Cell Density	Agonist (Concentration)	Selection Antibiotic
THP1-XBlue™-MD2-CD14	5 × 10^5^ cells/mL	LPS-EK Ultrapure (10 ng/mL)	Zeocin (200 µg/mL) G418 (250 µg/mL)
HEK-Blue™ hTLR2	2.8 × 10^5^ cells/mL	Pam3CSK4 (100 ng/mL) hTLR2-TLR1 heterodimer	HEK-Blue selection™
HEK-Blue™ hTLR4	1.4 × 10^5^ cells/mL	LPS-EK Ultrapure (10 ng/mL)	HEK-Blue selection™

## Data Availability

All the data used in the current study are available from the corresponding author upon reasonable request.

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
