# Peer review of "Swine Gut Lactic Acid Bacteria and Their Exopolysaccharides Differentially Modulate Toll-like Receptor Signaling Depending on the Agave Fructans Used as a Carbon Source"

_animals, 2025, doi:10.3390/ani15071047_

Round 1
Reviewer 1 Report
Comments and Suggestions for Authors
In general, the article approach a topic of interest and edge research. It also presents the results in an organized manner, reply the questions raised by each experiment. In the discussion, the results are compared with those obtained by other working groups, supporting their results.
Regarding specific comments, I only have two observations: the introduction fails to mention the importance of this type of research in terms of its impact on pig production and, consequently, the economic benefits it represents. The materials and methods section fails to mention the certification and standards that validate the management of these organisms.
Author Response
Dear Reviewer:
We sincerely appreciate your time and effort in evaluating our manuscript. Your insightful comments and constructive suggestions have been invaluable in enhancing the quality and clarity of our work. We have carefully considered each point raised and have made the necessary revisions to address all concerns.
Reviewer #1
Comments:
In general, the article approach a topic of interest and edge research. It also presents the results in an organized manner, reply the questions raised by each experiment.
In the discussion, the results are compared with those obtained by other working groups, supporting their results.
Regarding specific comments, I only have two observations: the introduction fails to mention the importance of this type of research in terms of its impact on pig production and, consequently, the economic benefits it represents. The materials and methods section fails to mention the certification and standards that validate the management of these organisms.
Response: Thank you for your observation; as you pointed out, lactic acid bacteria have been widely used as supplements in pig production, promoting weight gain and increasing average daily feed intake and feed-to-gain ratio. Specifically, LABs improve performance during pregnancy, parturition, and lactation in sows; they can enhance immunohematological parameters and defenses in the growing phase, influence the quality of meat in the finishing phase, and help reduce environmental pollutants. In terms of the economic consequences, such a prebiotics-probiotics system can reduce mortality and diminish the costs attributable to veterinary treatment, which produces benefits from the sale of live pigs. It has been demonstrated that piglets fed with growth promoters based on prebiotics and probiotics had better economic benefits. This addition increases weight gain and improves feed conversion, with a net economic benefit of 1:7.5 compared to animals without supplementation.
We added a paragraph concerning these observations to the Introduction section of the corrected manuscript.
Concerning the certification and standards that validate the management of swine, we added the following to the Materials and Methods section:
Ethical Statement
All the procedures were approved by the Institutional Research Committee and the Committee for the Care and Use of Laboratory Animals (IACUC-INP) under project register code INP 2021/026. The institutional Research Committee is certified by the Federal Commission for the Protection against Sanitary Risks (COFEPRIS-Mexico), register number 17 CI 09 003 109.
Animals
The minipigs (Sus scrofa domesticus) used in this study belong to a local genetic line developed by a breeder through selective crossbreeding of small pigs from various breeds. The primary selection included pigs with white skin and others with white coats featuring black or brown spots. At birth, their average weight is 250 g, and by the age of two years, they reach a body weight ranging between 25 and 35 kg. The minipigs were obtained from RGS Research Global Solutions (RGS07090347A), located at Av Paseo de la Reforma No 50 G, 401, Colonia Morelos, CDMX, C.P. 06200. Since the establishment of the colony in 2015, all animals have been housed in a purpose-built, conventionally managed, and ventilated facility specifically designed for minipigs. The facility complies with the NOM 062 ZOO 1999 guidelines (https://www.gob.mx/cms/uploads/attachment/file/203498/NOM-062-ZOO- 1999_220801.pdf). The location ensures biosecurity, as no other pig farms are within a 20 km radius. Additionally, a dedicated caretaker is responsible for all animals, does not work with other swine populations, and follows strict protective measures, including specialized clothing when entering the unit.
The animals were provided with untreated tap water and received a restricted diet consisting of PURINA feed (https://www.llabana.com/granja-familiar-finalizador). The feeding regimen included 200 g per meal, twice daily, at 9:00 AM and 5:00 PM, with feed placed directly on the floor. Monitoring sampling was done once a year (June 2024), and serum samples were taken for virus and bacterial antibodies from five animals. Samples were also taken for parasitological examination. Animals were free of Aujeszky, Circovirus, Epidemic Diarrhea of Pig, PRRS, Actinobacillus pleuropneumoniae, Pasterella spp., Bordetella bronchiseptica, Coccidia (Eimeria, Isospora) and Helminths. The animals underwent a one-month acclimatization period in the animal facility, adhering to the institution's internal regulations. The facility meets the standards outlined in the Guide for the Care and Use of Laboratory Animals (National Research Council) and is AAALAC-accredited, ensuring compliance with NOM062- ZOO-1999 regulations. The housing conditions included pens with slotted concrete floors, nipple drinkers, a controlled temperature range of 22–26°C, and relative humidity of 50% ± 20%. The pen sizes were designed to meet the space requirements for the minipigs, providing at least 1.08 m2 per animal. Each pig was housed in an individual pen measuring 1.12 m x 1.82 m.
To promote well-being and behavioral enrichment, the minipigs received daily socialization and participated in a standardized enrichment program. Their health and welfare were monitored twice daily to ensure optimal conditions throughout the study.
We thank the reviewer for his/her comments, which improved our manuscript. We hope the revisions made are satisfactory.

Reviewer 2 Report
Comments and Suggestions for Authors
1 Fecal samples containing LAB were just collected from five minipigs, what's the sampled population of the total minpig herd? and the differences on pig sex?
2 In 3.4, 3.4. LAB activate the NF-kB pathway can be actiated by LAB or its produced EPS, so how to tell which activation with independently of the carbon source or EPS-dependent in the culturing systems?
3 In Fig. 4B, The weakest activator of the NF-kB pathway was ranging from 2.5 to 7.6-fold……,please check the folds, including other numbers describled in the results.
4 Choosing TLR2 orTLR4 cell lines for activation assasy by EPS or LAB, what's the difference between TLR2 and TLR4?
Author Response
Dear Reviewer:
We sincerely appreciate your time and effort in evaluating our manuscript. Your insightful comments and constructive suggestions have been invaluable in enhancing the quality and clarity of our work. We have carefully considered each point raised and have made the necessary revisions to address all concerns.
Reviewer #2
1. Fecal samples containing LAB were just collected from five minipigs, what's the sampled population of the total minpig herd? and the differences on pig sex?
Response: Thank you for this observation and question. We sampled the total number of individuals in the herd (5): four males and one female. However, we isolated one LAB strain from each individual except for one individual (male #1) from which two strains were isolated. We included the following table as supplementary material to clarify this concern (Suppl. Table S1). Also, this was stated in the Materials and Methods section of the corrected manuscript.
2 In 3.4, LAB activate the NF-kB pathway can be actiated by LAB or its produced EPS, so how do we tell which activation is independent of the carbon source or EPS-dependent in the culturing systems?
Response: Thank you for this question, which drove our attention to interpret our results further.
In light of our results, activating the NF-kB pathway through the whole LAB cell is likely not carbon source-dependent in the culturing system. THP1 cells (which express all the TLRs that usually express a mammal cell) are activated similarly and independently of the LAB cell's concentration (see Fig. 4A). On the other hand, the EPS isolated from these LAB clearly activate THP1 cells in a dose-dependent manner and differentially in magnitude, depending on the carbon source in the culturing system (see Fig. 4B). Additionally, the most substantial fold change of activation was with EPS from LAB cultured in the presence of GTF A (17 times higher than the highest activation obtained with LAB cells). This supports that the EPS activation of the NF-kB pathway strongly depends on the carbon source in the culturing system, whereas it does not for LAB. However, further studies must be done to understand the underlying mechanisms of such behavior.
We added this dissertation in the Discussion section of the corrected manuscript.
3 In Fig. 4B, The weakest activator of the NF-kB pathway was ranging from 2.5 to 7.6-fold……,please check the folds, including other numbers described in the results.
Response: We really appreciate your precise observation. Indeed, those numbers were written wrong (maybe tiredness was the cause). We reviewed all numbers to correct them accordingly.
4 Choosing TLR2 or TLR4 cell lines for activation assasy by EPS or LAB, what's the difference between TLR2 and TLR4?
Response: Although TLR2 and TLR4 activate the NF-kB pathway to comparable extents, they exhibit notable differences in their impact on cytokine and chemokine gene transcription. This disparity indicates that the signals generated by these TLRs are far from identical. Moreover, the distinct roles of their pattern recognition receptors may significantly influence the polarization of adaptive immunity, underscoring their importance in immune response modulation. Cytokine production through TLR4 activation is associated with type I immunity (Th1) responses, whereas TLR2 activation produces cytokines that favor Th2 development (Re, F., & Strominger, J. L. 2001). Th1 activates cell types that are actively involved in controlling infection by intracellular pathogens, and Th2 stimulates antibody production to control infections by extracellular pathogens (Murtaugh, M. P., et al.). Therefore, assaying activation with TLR2 and TLR4 cell lines helped us to explore the potential activation of the immunity type that could exert the different LAB and EPS conditions explored herein. Since detailed analyses of the produced cytokines are still lacking, we can not precisely conclude what kind of pathogens (intracellular or extracellular) are better targets for the LAB and EPS systems studied in this work.
We added the above in the Discussion section of the corrected manuscript.
We thank the reviewer for his/her comments, which improved our manuscript. We hope the revisions made are satisfactory.

Reviewer 3 Report
Comments and Suggestions for Authors
Line 77 – New nomenclature should be considered, because there are others genus involved.
Line 115 – maybe it´s important to know the nutrients of the Purina ham, does it have GTFs?
Line 145 – The paragraph is not clear enough. Were de growth curves in the culture media with different carbon source prepared from an overnight culture adjusted to 0,1?
Line 185 – Witch controls did you used to perform that assay?
Line 185 – I suggest more detail for the “Assays with reporter cell lines” part in material and methods. It will help to understand better the results. For example, it seems something is missing when you read at line 242.
Line 189 – Write in parentheses what the letters of SEAP means.
Lines 225 and 226 – New nomenclature must be used. The identification of the strains should be given if they are well known, or references.
In the whole manuscript, I suggest to specify and write the identification of the strains that you compared with (are the strains from another study?), and use the new nomenclature.
Line 377 – the strains should be specified, because the strains of a same species can have different effects among them. Do the strains have a name? What are the characteristics of those strains?
Author Response
Line 77 – New nomenclature should be considered, because there are others genus
involved.
Response: We agree. Since the Lactobacillus genus has been replaced by a
considerably high number of new genera, and we were referring to a general idea of
probiotics in swine; therefore, the sentence at this line was changed as follows:
Various bacterial genera have been used as probiotics in swine; however, acid lactic
bacteria (LAB) members are the most widely used probiotic agents.
Line 115 – maybe it´s important to know the nutrients of the Purina ham, does it
have GTFs?
Response: The Purina ham does not contain Agave GTFs. However, it contains 6.5
% of fibers composed of subproducts of cereals such as wheat bran, corn bran, and
cane molasses. On the other hand, it is known that GTFs are present in wheat grains
(Verspreet , et al., 2015). Based on this information, we speculate that the
composition of fibers in the Purina ham might have favored the LAB grown.
However, once the LAB was isolated, it was only grown under Agave GTFs as a
carbon source. Thus, we inferred that the Purina ham does not influence the results
of this study.
Nevertheless, based on your comment, we added this information in the Materials
and Methods section.
Line 145 – The paragraph is not clear enough. Were the growth curves in the culture
media with different carbon sources prepared from an overnight culture adjusted to
0,1?
Response: Thank you for your observation. Yes, the growth curves in the culture
media with different carbon sources prepared from an overnight culture were
adjusted to 0.1. To clarify, we have rephrased the paragraph as follows: For strain
inoculation, the number of cells was adjusted to a final OD 600 nm of 0.1.
Line 185 – Witch controls did you used to perform that assay?
Response: Positive controls were THP1 cells challenged with 10 ng/mL Escherichia
coli lipopolysaccharide (LPS), HEK-Blue TLR4 cells also challenged with 10 ng/mL
E.coli LPS, and HEK-Blue TLR2 cells challenged with 100 ng/mL triacylated
lipopeptide Pam3CSK4. THP1 cells cultured only with RPMI and HEK Blue cells
cultured only with DMEM medium were considered negative controls.
Also, considering the following comment, this information was included in the
Methods section of the corrected manuscript.
Line 185 – I suggest more detail for the “Assays with reporter cell lines” part in
material and methods. It will help to understand better the results. For example, it
seems something is missing when you read at line 242.
Response: Thank you for your suggestion. To clarify the assays with reporter cell
lines, we have rewritten this section as follows: This cell line is derived from the
human THP1 monocyte cell line, which endogenously expresses all human TLRs. It
has been genetically modified for the stable integration of an NF-κB-inducible SEAP
(secreted embryonic alkaline phosphatase) reporter construct. Thus, when
environmental stimuli activate TLRs, the activation of the NF-κB pathway can be
quantified through SEAP production, which can be spectrophotometrically
registered at 655 nm. Cells cultured only with RPMI medium were considered as
negative controls. Positive controls were cells challenged with 10 ng/mL Escherichia
coli lipopolysaccharide (LPS) (Table 1). Besides, the human embryonic kidney cells
(HEK-Blueä) that independently express TLR2 and TLR4 were used. These cell lines
are also genetically modified for the SEAP production induced by the NF-κB
pathway promoter. The cell densities, agonists used as positive controls, and
selection antibiotics are shown in Table 1.
THP1 and HEK-Blue cell lines were cultured in RPMI-1640 and DMEM medium,
respectively. Both media were supplemented with 10% deactivated fetal bovine
serum, 100 µg/ml normocin, and penicillin/streptomycin 50 U/mL and 50 µg/ml,
respectively. According to manufacturer instructions, cell lines were passaged twice
weekly and used at 80% confluency.
Line 189 – Write in parentheses what the lelers of SEAP means.
Response: It means secreted embryonic alkaline phosphatase. As mentioned above,
it was included in the corrected manuscript version.
Lines 225 and 226 – New nomenclature must be used. The identification of the
strains should be given if they are well known, or references.
Response: Thank you for the observation. We have changed Lactobacillus reuteri to
Limosilactobacillus reuteri, and Lactobacillus casei to Lacticaseibacillus casei. We changed
the nomenclature in the corrected version of the manuscript.
Additionally, in the methods section, we have identified the strains based on the
ATCC code as a reference. The Limosilactobacillus reuteri strain is F275 [DSM 20016]
from ATCC 23272, and the Lacticaseibacillus casei strain is 03 from ATCC 393.
In the whole manuscript, I suggest to specify and write the identification of the
strains that you compared with (are the strains from another study?), and use the
new nomenclature.
Response: We agree. The strains used to compare were those mentioned above, and
we identified them in the whole corrected manuscript. These strains are available in
our Research Institute and have been used for other studies.
Line 377 – the strains should be specified, because the strains of a same species can
have different effects among them. Do the strains have a name? What are the
characteristics of those strains?
Response: The strain of Bifidobacterium longum subsp. infantis was DSM 20088 (Also
found in other collections with the codes ATCC 15697, NCTC 11817, and CCUG
18368). It was isolated from a human infant intestine. This strain metabolizes
arabinose, xylose, nannose, and trehalose. Its cell wall peptidoglycan is of the type
L-Orn-L-Ser-L-Ala-Thr-L-Ala
The strain of Bifidobacterium adolescentis was DSM 20083 (Also found in other
collections with the codes ATCC 15703, NCTC 11814, and CCUG 18363). It was
isolated from a human adult intestine. This strain metabolizes arabinose, xylose,
salicin, mannitol, trehalose, and cellobiose. Its cell wall peptidoglycan is of the type
L-Orn-(L-Lys)-D-Asp + L-Lys-L-Ser-(L-Ala)2.
We specified the strains in the corrected manuscript and included the reference
where the reader can find their detailed characteristics.
We thank the reviewer for his/her comments, which improved our manuscript. We
hope the revisions made are satisfactory.

Reviewer 4 Report
Comments and Suggestions for Authors
The article of interest, touches on the points of influence of EPS synthesis under different carbon source conditions.
However, there are comments
1. In the Materials and methods part it is not described where the strains Lactobacillus cassei, and L. reuteri were obtained from, the authors use these strains in the work.
2 The authors did not explain why these particular strains of Lactobacillus cassei, and L. reuteri were chosen as a comparison.
3. The authors did not describe the technique of LAB activate the NF-B, whereas this is given much attention in the results
4. There is no description of the methodology of the NF-B pathway in THP1 reporter cells.
5. Decode LPS and other abbreviations
I would like to point out the good discussion of the results.
Comments on the Quality of English LanguageI'm no expert in English
Author Response
Reviewer #4
The article of interest, touches on the points of influence of EPS synthesis under different carbon source conditions. However, there are comments.
- In the Materials and methods part it is not described where the strains Lactobacillus cassei, and L. reuteri were obtained from, the authors use these strains in the work.
Response: Thank you for the observation. Based on the comments of another reviewer, we actualized the nomenclature to refer to these strains. We have changed Lactobacillus reuteri to Limosilactobacillus reuteri, and Lactobacillus casei to Lacticaseibacillus casei. We have already changed the nomenclature in the corrected version of the manuscript.
Additionally, in the methods section, we have identified the strains based on the ATCC code as a reference. The Limosilactobacillus reuteri strain is F275 [DSM 20016] from ATCC 23272, and the Lacticaseibacillus casei strain is 03 from ATCC 393.
2 The authors did not explain why these particular strains of Lactobacillus cassei, and L. reuteri were chosen as a comparison.
Response: Currently, these strains are available at our research institute and have been used for other studies. Then, the facility to obtain these strains that have been primarily characterized was the main reason for using them for comparisons. We hope this doesn't matter, but sometimes, economic resources are scarce.
- The authors did not describe the technique of LAB activate the NF-kB, whereas this is given much attention in the results.
Response: Thank you for this observation.
- There is no description of the methodology of the NF-kB pathway in THP1 reporter cells.
Response: We have rewritten this section as follows: THP1 is a cell line derived from the human THP1 monocyte cell line, which endogenously expresses all human TLRs. It has been genetically modified for the stable integration of an NF-κB-inducible SEAP (secreted embryonic alkaline phosphatase) reporter construct. Thus, when environmental stimuli activate TLRs, the activation of the NF-κB pathway can be quantified through SEAP production, which can be spectrophotometrically registered at 655 nm. Cells cultured only with RPMI medium were considered as negative controls. Positive controls were cells challenged with 10 ng/mL Escherichia coli lipopolysaccharide (LPS) (Table 1).
- Decode LPS and other abbreviations.
Response: Done.
I would like to point out the good discussion of the results.
We thank the reviewer for his/her comments, which improved our manuscript. We are flattered to read your opinion about our discussion of the results. We hope the revisions made are satisfactory.

Round 2
Reviewer 3 Report
Comments and Suggestions for Authors
The manuscript is clearer now and well written and understandable. It's an important research and reflects a lot of work.
Reviewer 4 Report
Comments and Suggestions for Authors
The paper was improved
Comments on the Quality of English LanguageI'm not expert for english